# The presynaptic glycine transporter GlyT2 is regulated by the Hedgehog pathway in vitro and in vivo

Andrés de la Rocha-Muñoz[1,2,4], Enrique Núñez[1,2,4], Anjali Amrapali Vishwanath[3], Sergio Gómez-López [1],
Dhanasak Dhanasobhon[3], Nelson Rebola[3], Beatriz López-Corcuera[1,2], Jaime de Juan-Sanz [3✉] &
Carmen Aragón[1,2]

The identity of a glycinergic synapse is maintained presynaptically by the activity of a surface glycine transporter, GlyT2, which recaptures glycine back to presynaptic terminals to preserve vesicular glycine content. GlyT2 loss-of-function mutations cause Hyperekplexia, a rare neurological disease in which loss of glycinergic neurotransmission causes generalized stiffness and strong motor alterations. However, the molecular underpinnings controlling GlyT2 activity remain poorly understood. In this work, we identify the Hedgehog pathway as a robust controller of GlyT2 expression and transport activity. Modulating the activation state of the Hedgehog pathway in vitro in rodent primary spinal cord neurons or in vivo in zebrafish embryos induced a selective control in GlyT2 expression, regulating GlyT2 transport activity. Our results indicate that activation of Hedgehog reduces GlyT2 expression by increasing its ubiquitination and degradation. This work describes a new molecular link between the Hedgehog signaling pathway and presynaptic glycine availability.

[1] Centro de Biología Molecular "Severo Ochoa", Universidad Autónoma de Madrid, Consejo Superior de Investigaciones Científicas, 28049 Madrid, Spain.
[2] IdiPAZ, Hospital Universitario La Paz, Madrid, Spain. [3] Sorbonne Université, Institut du Cerveau - Paris Brain Institute - ICM, Inserm, CNRS, APHP, Hôpital de la Pitié Salpêtrière, Paris, France. [4] These authors contributed equally: Andrés de la Rocha-Muñoz, Enrique Núñez. ✉email: jaime.dejuansanz@icm-institute.org

Glycinergic neurotransmission plays a fundamental role in neuronal circuits of the central auditory pathway, receptive fields in the retina, and spinal cord-sensitive pathways[1]. Glycine availability in the cytosol of glycinergic terminals determines the rate and extent of synaptic vesicle filling, controlling the size of the quantum of transmission[2–5]. However, presynaptic glycine concentration is not maintained by local synthesis but relies on robust glycine recapture by the surface glycine transporter GlyT2[3], a transporter of the solute carrier family (SLC) family[1]. Alterations in GlyT2 expression or activity considerably reduce glycine content in synaptic vesicles, vastly weakening glycinergic neurotransmission[3,4,6]. In humans, this dysfunction is the main presynaptic cause of hyperekplexia[7–10], but it also may be involved in the pathology of chronic pain[11] and deficits in auditory processing[12]. A better understanding of the molecular regulation of GlyT2 would help framing future studies on the importance of this transporter in human disease and we and others have identified some molecular mechanisms controlling GlyT2 expression and activity[13–23]. Ubiquitination, a post-translational modification in which the small protein ubiquitin is covalently attached to a cytoplasmic lysine residue of a protein, is a major control point that finely tunes the expression of GlyT2[14,16,24,25]. This mechanism is shared with other neurotransmitter transporters[26–32] although the E3 ubiquitin ligases involved in the process may differ in each case[24,33,34].

GlyT2 expression is necessary for establishing and maintaining the identity of a glycinergic synapse[3,35–37]. During development, GlyT2 expression in mice increases during synaptogenesis and precedes the appearance of the α1 subunit isoform of the adult glycine receptor (GlyR)[38,39], which mediates the activity switch of the GlyR from depolarizing to hyperpolarizing that it is required for a functional inhibitory glycinergic synapse[36]. Studies in zebrafish also described that GlyT2 appears early in development at 20–24 h post-fertilization (hpf) and presents its maximum expression at 4–5 days post-fertilization in the dorsal region, when primary spinal cord neurogenesis appears complete[40,41]. These data suggest that GlyT2 may have an early role in establishing the identity of inhibitory glycinergic synapses[38] in addition to its well-established role in sustaining glycinergic neurotransmission strength in the mature central nervous system (CNS)[3,35–37]. Although GlyT2 developmental expression pattern is relatively well described and some transcription factors have been involved in its early appearance in spinal cord[42,43], the signaling pathways that may control this process remain unknown.

The Hedgehog signaling pathway is one of the intricate transduction mechanisms involved in embryonic development[44]. The pathway is activated when Hedgehog ligands bind to its surface receptor Patched (PTCH), promoting the derepression of Smoothened (SMO), a G-protein-coupled receptor that is inhibited by PTCH in basal conditions. The activation of SMO triggers a cascade of cellular events that ultimately induce the activation of the zinc-finger transcription factors called glioma-associated oncogene family members (GLI)[45,46]. In mammals, there are three Hedgehog ligands: Desert Hedgehog (Dhh), Indian Hedgehog (Ihh), and Sonic Hedgehog (Shh). Although they have similar affinities to bind PTCH and regulate Hedgehog target genes, they present distinct roles during development as consequence of their different expression patterns[47]. Dhh is involved in germ cell development and myelination of peripheral nerves, Ihh is needed for the development of bone and cartilage, and Shh is crucial for the developing brain of vertebrates as it controls the differentiation of neuronal populations, among other functions[48–52]. In addition, recent research shows that Shh also presents a role in the mature CNS where it may modulate the function of CNS neuronal circuits and synaptic activity[44,53–57].

Here we explored whether the Hedgehog pathway could control GlyT2 expression and activity. We discovered that glycinergic neurons respond to Hedgehog signaling both in vitro in rat brainstem and spinal cord primary neurons and in vivo in zebrafish embryos. Our data indicate that the Hedgehog pathway controls GlyT2 ubiquitination status, modulating its functional expression and transport activity. The control of presynaptic glycine availability by the Hedgehog pathway suggests that its activation may act as a regulatory mechanism that modulates inhibitory glycinergic neurotransmission strength, a crucial role in the establishment of the identity of functional glycinergic synapses in vertebrates.

## Results

**Hedgehog activation downregulates GlyT2 expression and transport activity.** To explore whether the Hedgehog pathway can modulate GlyT2, we used primary cultures of brainstem and spinal cord neurons where the transporter is highly expressed. First, we tested the effect of the Hedgehog agonist purmorphamine (PMM), a purine derivative that directly binds to and activates the G-protein coupled receptor SMO[58]. SMO activation is required for transmembrane signaling to the cytoplasm, ultimately resulting in the activation of GLI transcription factors[45]. As Fig. 1 shows, 10 μM of PMM caused a time-dependent inhibition of GlyT2, reducing glycine transport activity by 50% after 16 h of treatment (Fig. 1a, $n = 11$, ****$p < 0.0001$). We discovered that this effect is accompanied by a parallel decrease of 60% in GlyT2 protein levels (Fig. 1b, c; $n = 10$; ****$p < 0.0001$), suggesting that the impact in GlyT2 function observed during SMO activation is a direct consequence of the reduction of the number of active transporters in the surface of neurons. To investigate whether PMM induces selective GlyT2 protein decrease in brainstem and spinal cord neurons, we studied its effect on other proteins involved in glycinergic function. Activation of the Hedgehog pathway for 8 or 24 h did not alter levels of known GlyT2 interactors (Fig. 1d, e), such as the α3 subunit Na$^+$/K$^+$-ATPase (α3NKA)[22], CRMP5/Ulip6[18], or syntaxin1A (STX1A)[19]. This suggests that Hedgehog activation selectively controls GlyT2 levels, leaving other functions of glycinergic neurons unaffected. We next carried out a dose–response analysis on the PMM effect on GlyT2 function and protein expression (Fig. 2). The results show that PMM inhibits [$^3$H]-glycine uptake (Fig. 2a, $n = 10$; Veh vs PMM 5 μM, *$p = 0.0242$; Veh vs PMM 10 μM, ****$p < 0,0001$) and decreases GlyT2 expression in a concentration-dependent manner after 16 h of treatment (Fig. 2b, $n = 4$; Veh vs PMM 5 μM, *$p = 0.0248$; Veh vs PMM 10 μM, **$p = 0.0098$). To corroborate that these effects and those shown in Fig. 1 are consequence of activating the Hedgehog pathway, we confirmed that the Hedgehog antagonist cyclopamine, a plant-derived steroidal alkaloid known to specifically inhibit SMO[59,60], inhibited the effect of PMM on GlyT2 (Fig. 2b, c; $n = 4$; Veh vs PMM 1 μM + CYC 0.2 μM, $^{ns}p > 0.9999$; Veh vs PMM 5 μM + CYC 1 μM, $^{ns}p > 0.9999$; Veh vs PMM 10 μM + CYC 2 μM, $^{ns}p > 0.9999$).

To better understand the underlying molecular mechanism driving a reduction in GlyT2 levels during PMM treatment, we analyzed whether Smo activation could regulate GlyT2 function by acting on its mRNA levels. Using qPCR, however, we did not detect any alteration in GlyT2 mRNA expression after 16 h of treatment (Fig. 3a, $n = 7$, $^{ns}p = 0.1868$), a condition that reduces both activity and protein levels by more than 50%, suggesting that PMM does not induce changes at the transcriptional or post-transcriptional levels to control GlyT2 expression. Given that ubiquitination is an essential post-translational modification that regulates GlyT2 expression and transport activity[14,24,25] and other neurotransmitter

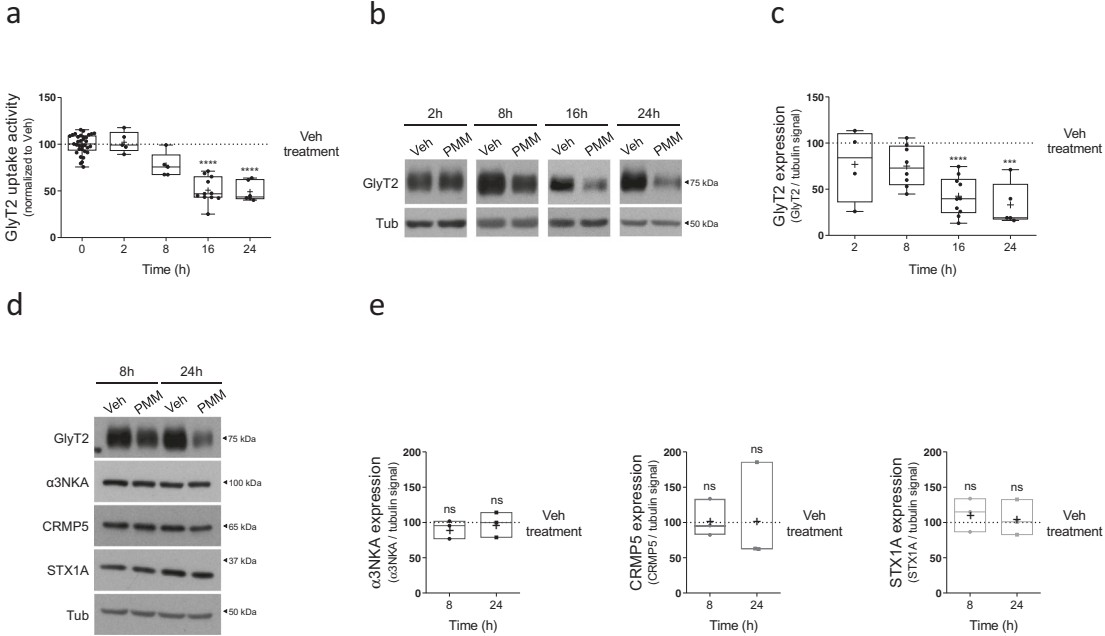

**Fig. 1 Purmorphamine treatment controls the functional expression of GlyT2 in a specific and time-dependent manner. a** Primary brainstem and spinal cord neurons were treated with 10 μM purmorphamine at the indicated times and glycine transport rates were measured using [$^3$H]-glycine transport assays. Glycine transport shown is normalized against control conditions. ****$p$ (PMM 16 h) < 0.0001, ****$p$ (PMM 24 h) < 0.0001, using Dunn's multiple comparisons test, Number of experiments $n$ (Veh & PMM 2 h) = 5, $n$ = (Veh & PMM 8 h) = 5, $n$ (Veh & PMM 16 h) = 11, $n$ (Veh & PMM 24 h) = 6. **b** Representative immunoblot of primary brainstem and spinal cord neuronal cultures. Cells were treated with 10 μM purmorphamine for the times indicated. Tubulin is used as protein loading control. Veh: DMSO. **c** Quantification is shown normalized to the corrected signal in the control in each case (Veh). ****$p$ (PMM 16 h) < 0.0001, ***$p$ (PMM 24 h) = 0.0002, using Dunn's multiple comparisons test, number of experiments $n$ (Veh & PMM 2 h) = 4, $n$ = (Veh & PMM 8 h) = 8, $n$ (Veh & PMM 16 h) = 10, $n$ (Veh & PMM 24 h) = 5. **d** Neurons were treated with the vehicle alone or with 10 μM purmorphamine and the expression of the following GlyT2-associated/related proteins was assessed: α3NKA, CRMP5, and syntaxin1A (STX1A). Tubulin is used as protein loading control. **e** Box plots show quantification of protein expression changes during 8 and 16 h PMM treatment measured by western blots performed as in Fig. 1d. Data are normalized to the corrected signal in the control in each case (Veh). $^{ns}p$ (αNKA: PMM 8 h) = 0.7260, $^{ns}p$ (αNKA: PMM 24 h) > 0.9999, $^{ns}p$ (CRMP5: PMM 8 h) > 0.9999, $^{ns}p$ (CRMP5: PMM 24 h) > 0.9999, $^{ns}p$ (STX1A: PMM 8 h) = 0.7260, $^{ns}p$ (STX1A: PMM 24 h) > 0.9999, using Dunn's multiple comparisons test, number of experiments $n$ = 3. PMM purmorphamine.

transporters[29–31,61], we decided to explore whether this process is involved in the downregulation of GlyT2 by PMM. We carried out established ubiquitination assays[24,62] in primary neurons treated for 16 h with PMM 10 μM or vehicle (Fig. 3b, c). In these assays, GlyT2 is immunoprecipitated under very stringent conditions from samples incubated at 95 °C for 10 min in a buffer containing 1% sodium dodecyl sulfate (SDS). This treatment denatures all proteins, disassembling their native structure and completely disrupting non-covalent protein–protein interactions. However, given that ubiquitin molecules are covalently attached to GlyT2, this treatment does not affect the number of ubiquitin molecules bound to GlyT2, allowing to quantify GlyT2 ubiquitination by immunoblotting GlyT2 immunoprecipitates against ubiquitin. As expected, given that the anti-ubiquitin antibody used here recognizes all forms of polyubiquitin chains, which are formed by variable chain lengths, these experiments identify an elongated band spanning molecular weights higher than GlyT2, which correspond to immunoprecipitated GlyT2 molecules ubiquitinated by variable polyubiquitin lengths. To control for variability on GlyT2 expression and immunoprecipitation, the ubiquitin signal is normalized against the immunoprecipitated amount of GlyT2 in each case. Compared to the control (Fig. 3b, Veh), an increased ubiquitination signal was found when neurons were treated with PMM (Fig. 3b, PMM), showing that indeed the Hh agonist promotes GlyT2 ubiquitination. Note the robust reduction of GlyT2 expression levels by PMM treatment in these experiments (Fig. 3b, lower blot) in agreement with Figs. 1b, c and 2b, c. Quantification of these data indicated that agonist treatment significantly increased GlyT2 ubiquitination levels

by ~2.5-fold (Fig. 3c, $n$ = 3, **$p$ = 0.0074). To confirm that ubiquitination of GlyT2 is the mechanism underlying the down-regulation observed when the Hedgehog pathway is activated by PMM, we used PYR41, a cell-permeable specific inhibitor of the E1 ubiquitin-activating enzyme that catalyzes an initial and critical step in the protein ubiquitination pathway[63]. As Fig. 3d, e shows, when neurons were treated with PMM in the presence of PYR41 we no longer observed GlyT2 downregulation ($n$ = 4; Veh vs PMM, *$p$ = 0.0164; Veh vs PMM + PYR41, $^{ns}p$ > 0.9999), suggesting that indeed PMM-mediated control of GlyT2 expression requires ubiquitination of the transporter.

We recently identified LNX2 as the first E3 ubiquitin ligase controlling GlyT2 ubiquitination and expression levels in neurons[24]. Thus, we wondered whether the control of GlyT2 by the Hedgehog pathway requires the function of LNX2. To do this, we ablated the expression of LNX2 using a viral knockdown approach in primary neurons as previously described[24]. In these conditions we observed an increase in the expression of GlyT2 due to the lack of constitutive ubiquitination of the transporter, as previously reported[24]. However, the genetic deletion of endogenous LNX2 in spinal cord primary neurons neither abolished the effect of PMM in GlyT2 activity (Fig. S1a, $n$ = 4; PMM effect: Control shRNA vs LNX2 shRNA, $^{ns}p$ = 0.6814) nor expression (Fig. S1b, c, $n$ (Control) = 3, $n$ (shRNA LNX2) = 7; PMM effect: Control shRNA vs LNX2 shRNA, $^{ns}p$ = 0.9401) indicating that ubiquitination of GlyT2 by activation of the Hedgehog pathway requires other E3 ligases to mediate this effect. The qPCR analysis of the LNX2 mRNA levels in neurons treated with PMM showed

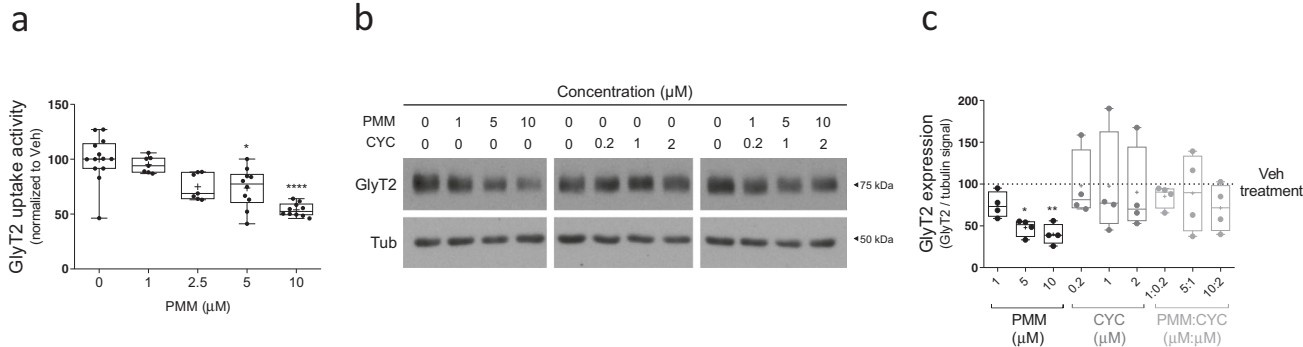

**Fig. 2 Downregulation of GlyT2 by purmorphamine is concentration-dependent and is reverted through cyclopamine. a** Primary brainstem and spinal cord neurons were treated with the concentrations of purmorphamine indicated for 16 h and glycine transport rates were measured using [$^3$H]-glycine transport assays. Glycine transport shown is normalized against control conditions. *$p$ (PMM 5 µM) = 0.0242, ****$p$ (PMM 10 µM) < 0.0001, using Dunn's multiple comparisons test, $n$ (Veh) = 13, number of experiments $n$ (PMM 1 µM) = 7, $n$ (PMM 2.5 µM) = 7, $n$ (PMM 5 µM) = 10, $n$ (PMM 10 µM) = 10. **b** Representative immunoblot of primary brainstem and spinal cord neurons. Cells were treated with purmorphamine or/and cyclopamine at the concentrations indicated and the expression of GlyT2 was analyzed by immunoblotting after 16 h. Tubulin is used as protein loading control. **c** Quantification of GlyT2 expression is normalized to the corrected signal against tubulin. *$p$ (Veh vs PMM 5 µM) = 0.0248, **$p$ (Veh vs PMM 10 µM) = 0.0098, $^{ns}p$ (Veh vs PMM 1 µM + CYC 0.2 µM) > 0.9999, $^{ns}p$ (Veh vs PMM 5 µM + CYC 1 µM) > 0.9999, $^{ns}p$ (Veh vs PMM 10 µM + CYC 2 µM) > 0.9999, using Dunn's multiple comparisons test; number of experiments $n$ = 4. PMM purmorphamine, CYC cyclopamine.

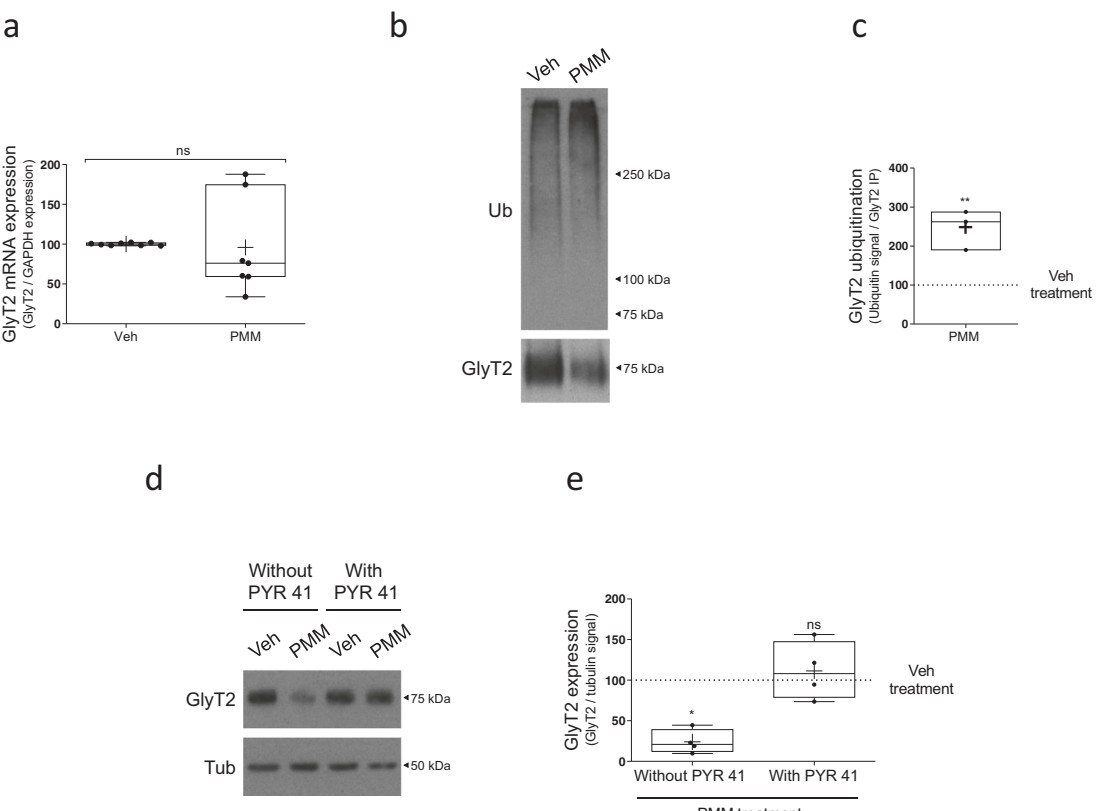

**Fig. 3 The ubiquitination of GlyT2 is the underlying mechanism of purmorphamine modulation. a** Neurons were treated with 10 µM purmorphamine or vehicle for 16 h. Relative GlyT2 mRNA levels were determined by qPCR using glyceraldehyde-3-phosphate dehydrogenase (GAPDH) as housekeeping gene (arbitrary units). ns = not significantly different using Mann–Whitney test, $n$ = 7. **b** Neurons were treated with purmorphamine as described in **a**. GlyT2 was immunoprecipitated and ubiquitination of the transporter was assayed by immunoblotting with anti-ubiquitin antibody. Blots were probed against GlyT2 to normalize ubiquitination signal against the amount of GlyT2 immunoprecipitated in each case to correct for GlyT2 protein expression. **c** Quantification of GlyT2 ubiquitination normalized to the control (vehicle). **$p$ = 0.0074, using unpaired $t$-test, $n$ = 3. **d** Neurons were treated with purmorphamine as described in **a**. and in the presence or absence of 5 µM PYR41. Tubulin is used as protein loading control. **e** Quantification of GlyT2 expression was normalized against tubulin. *$p$ (Veh vs PMM) = 0.0164, $^{ns}p$ (Veh vs PMM + PYR41) > 0.9999 using Kruskal–Wallis test, $n$ = 4. PMM purmorphamine.

no differences with respect to the control situation (Fig. S1d, $n = 3$, $^{ns}p = 0.3964$), suggesting that the lack of involvement of LNX2 in Hedgehog-mediated control of GlyT2 is not due to a decrease in its expression.

Considering that PKC can be activated by Hedgehog pathway[64] and that activation of PKC in glycinergic neurons results in an increased ubiquitination of GlyT2 and reduction of its expression and transport activity[13,14,25], we studied to what extent PKC activation is involved in the control of GlyT2 by the Hedgehog pathway. We treated primary neurons with the Hedgehog agonist together with bisindolylmaleimide, a selective PKC antagonist, and measured GlyT2 expression. We found that GlyT2 decrease was still observed despite PKC inhibition using a set of concentrations (Fig. S2a, b, $n = 4$; Control: Veh vs PMM, $^{**}p = 0.0020$; Bis 20 μM: Veh vs PMM, $^*p = 0.0491$; Bis 100 μM: Veh vs PMM, $^{**}p = 0.0018$; Bis 500 μM: Veh vs PMM, $^*p = 0.0491$) suggesting that GlyT2 reduction in these conditions occurs in a PKC-independent manner. Considering that both LNX2 and PKC modulate GlyT2 expression and function[14,24,25], the apparent independence of these pathways in the mechanism underlying GlyT2 downregulation by PMM identified in this work suggests that other downstream targets of the Hedgehog pathway are involved.

**Hedgehog inhibition in vivo in zebrafish embryos increases GlyT2 expression.** In order to investigate whether Hedgehog-mediated control of GlyT2 expression in primary neurons of rat brainstem and spinal cord is also conserved in vivo, we next used zebrafish (*Danio rerio*) embryos, a useful model to study CNS development, molecular mechanisms of neuropathology, or behavioral neuroscience[65–67]. When testing the effect of a drug in the zebrafish CNS, a common practice is to use dechorionated embryos before 72 hpf, time at which the blood–brain barrier is fully formed, to ensure the compound reaches its neuronal targets[68–70]. GlyT2 expression in zebrafish embryos appears early at 20–24 h post-fertilization (hpf), increases over time, and is clearly detected at 36–48 hpf in dorsal interneurons[40,41,71]. We first treated dechorionated zebrafish embryos with the agonist of SMO, PMM, at 24 hpf and using different concentrations (10, 25, and 50 μM). However, we did not detect a reduction in GlyT2 (Fig. S3). This may not be surprising, as at this developmental stage, the Shh signaling pathway is already robustly activated in zebrafish and pharmacological overstimulation on top may not be easily achievable. However, it is well described that the inhibition or ablation of Shh pathway causes molecular and developmental defects in the dorsoventral axis of the zebrafish neural tube[72–77]. Thus, we next decided to treat dechorionated zebrafish embryos with cyclopamine (CYC), antagonist of SMO, to study the control of GlyT2 expression by the Hedgehog pathway in vivo. We found a significant increase in GlyT2 expression in zebrafish treated with cyclopamine for 24 h (Fig. 4a, b, $n = 8$, 4 zebrafish per condition in each experiment; $^{**}p = 0,0014$), suggesting that at this stage of the zebrafish embryonic development GlyT2 is constitutively being downregulated in vivo through the physiological activation of the Hedgehog pathway. We also quantified GlyT2 mRNA in these conditions, but our qPCR analysis showed no change in GlyT2 mRNA levels (Fig. 4c, $n = 4$, 10 zebrafish per condition in each experiment; $^{ns}p = 0.3429$). To further validate these results, we used immunofluorescence confocal microscopy to visualize the effect of cyclopamine treatment on GlyT2 expression in the zebrafish spinal cord. Immunohistochemistry of zebrafish embryos revealed a significant increase in the GlyT2 expression upon cyclopamine treatment at 48 hpf compared to control (Fig. 4d, e, $n$ (Veh) = 29, $n$ (CYC) = 30; $^{****}p < 0.0001$), confirming western blot results. These in vivo data confirm our

initial in vitro results and suggest that the modulation of GlyT2 by the Hedgehog pathway is conserved among different vertebrates.

**Discussion**

GlyT2 function is essential to sustain glycinergic neurotransmission strength in vertebrates[3,4,6] and its dysfunction causes hyperekplexia, a rare disease characterized by hypertonia and pronounced startle responses to stimuli that can cause sudden infant death[6,36,78]. In this work, we identify a novel molecular link between the activation status of the Hedgehog pathway and the function of GlyT2. Our data reveal that the activation of the G-protein-coupled receptor SMO, mediator of Hedgehog pathway, induces an agonist concentration- and time-dependent decrease in GlyT2 transport activity and protein expression in primary neurons of rat brainstem and spinal cord and, interestingly, this mechanism affects GlyT2 selectively and does not modulate the expression of other proteins involved in glycinergic function. According to these results, pharmacological inhibition of SMO in vivo in zebrafish embryos promotes an increase in GlyT2 protein expression. These data provide the first in vitro and in vivo evidence of the control of GlyT2 function by the Hedgehog pathway

Experimentally modulating SMO function allowed us to discover that the Hedgehog pathway controls GlyT2 expression, but further work will be needed to dissect which of the three Hedgehog ligands (Dhh, Ihh, and Shh) endogenously controls GlyT2 levels. While these ligands have similar capacity to upregulate Hedgehog by binding PTCH[47,49] and causing SMO derepression, their expression patterns differ and so does their likelihood to modulate GlyT2. For example, Dhh expression is restricted to gonads and peripheral nerves[49,79] and therefore it is unlikely that it could act onto glycinergic neurons. Ihh, in turn, is involved in different processes in CNS[75,80], but it remains unclear to what extent it may function in the context of the active Shh signaling observed during neural tube development[80]. Lastly, Shh functions in CNS are well described and its role in the CNS has been classically associated to controlling brain and spinal cord development by driving cell proliferation, specification, and axonal targeting in multiple sites[52,54,81]. Based on the results obtained in this work and the requirement of GlyT2 expression for the establishment of a functional glycinergic synapse, it is tempting to speculate that Shh signaling could modulate GlyT2 expression in glycinergic neurons in the spinal cord to control the developmental timing of the beginning of spinal cord glycinergic transmission. Shh could provide a coordinated regulation of cellular differentiation and molecular specialization of glycinergic inhibitory neurons, facilitating the known function of the pathway as a key driver of neuronal diversity in the spinal cord[82]. In addition, glycine has been shown to be a regulator of neurogenesis in the spinal cord[83,84] and in a recent transcriptomic analysis performed in zebrafish embryos, Shh was identified as a modulator of glycine-dependent neurogenesis[85]. Although this could indicate that glycine signaling may control GlyT2 expression through Shh signaling during development and this could in turn be a homeostatic mechanism that regulates glycine levels to modulate neurogenesis, future work is needed to dissect this possibility.

On the other hand, recent results also indicate that Shh signaling is maintained after development and continues to act in the mature CNS, controlling synaptic physiology among other functions[44,50,54]. For example, the Hedgehog pathway has been shown to control excitatory synaptic function, as its signaling components are present in axonal terminals[55,86,87] and their activation increases release probability in excitatory

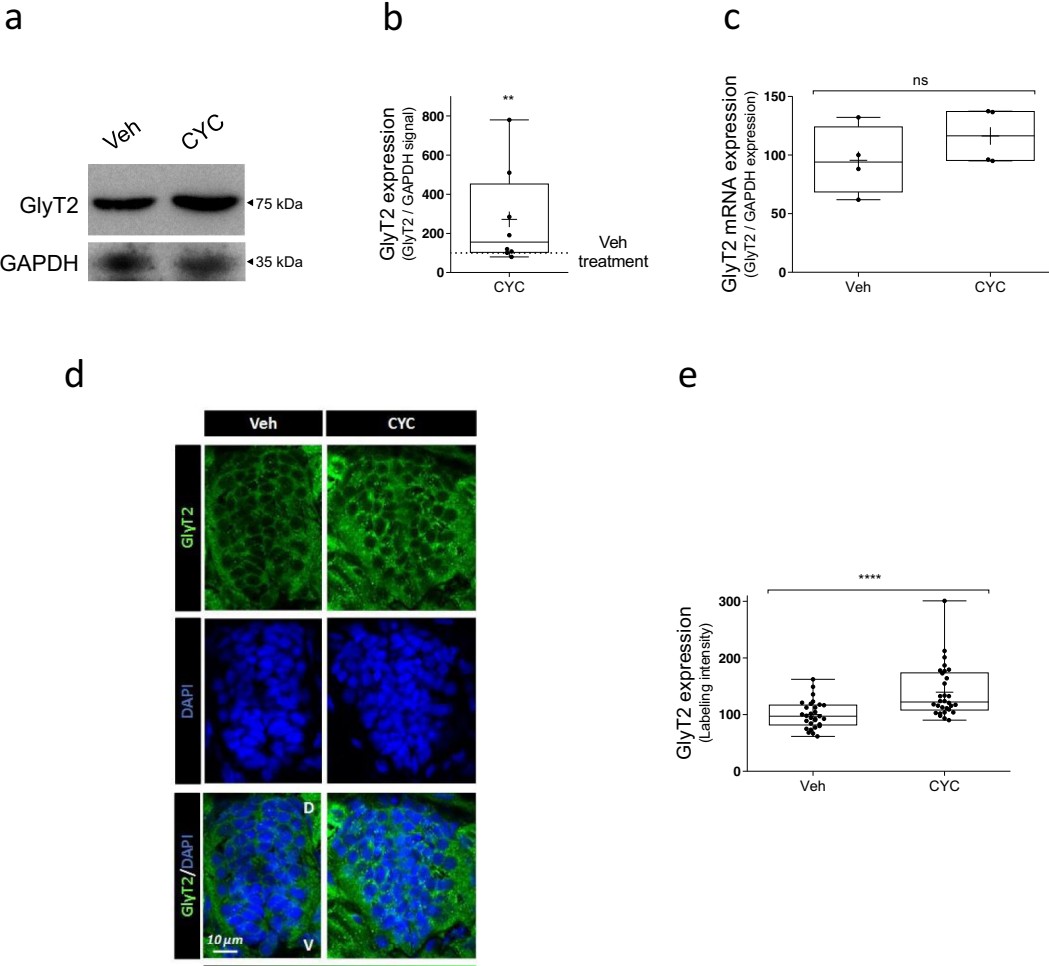

**Fig. 4 Inhibition of SMO receptor by cyclopamine upregulates GlyT2 expression in zebrafish embryos. a** Representative immunoblot of 48 hpf zebrafish embryos. Twenty-four hpf embryos were incubated with 100 μM cyclopamine or with vehicle alone (4 embryos per point) for 24 h and the expression of GlyT2 was analyzed by immunoblotting. GAPDH is used as protein loading control. **b** Quantification of GlyT2 expression is normalized to the corrected signal against GAPDH. **$p$ (Veh vs CYC 100 μM) = 0.0014, using Kolmogorov–Smirnov test, $n = 8$. **c** Relative mRNA levels of GlyT2 in 24 hpf zebrafish embryos treated with 100 μM cyclopamine for 24 h were determined by qPCR using glyceraldehyde-3-phosphate dehydrogenase (GAPDH) as housekeeping gene (arbitrary units). n.s. = not significantly different using Mann–Whitney $U$ test, $n = 4$. CYC cyclopamine. **d** Dechorionated 48 hpf zebrafish embryos treated as in **a**. were fixed in 4% PFA and embedded in OCT cryostat embedding medium. Ten micrometer spinal cord transversal sections were incubated with anti-GlyT2 antibody (green) and DAPI (blue). **e** Quantification of GlyT2 fluorescence intensity. ****$p < 0.0001$, using Mann–Whitney $U$ test, $n$ (Veh) = 29, $n$ (CYC) = 30. CYC cyclopamine, D dorsal, V ventral.

hippocampal neurons[55]. Shh additionally inhibits the activity of EAAC1 and GLT-1 glutamate transporters, causing a dramatic increase in extracellular glutamate levels that leads to imbalanced neuronal circuitry and seizures[53,88]. Although components of the Hedgehog pathway are expressed in the adult spinal cord in mice[52,89] it remains unexplored whether this pathway could be involved in shaping the function of established glycinergic synapses. Despite Shh has already been shown to modulate synaptic function of other neurotransmitters such as glutamate[53,55,88] or acetylcholine[90], whether Shh can regulate GlyT2 function to modulate glycinergic transmission will require future work. As GlyT2 is considered a main regulator of glycinergic quantal content, one could predict that modulating the activation state of the Hedgehog pathway could in turn modulate GlyT2 transport activity to the extent to which it could affect the amount of glycine released during neurotransmission. However, we did not address such possibility in this work and interpretations of the impact of Shh activity onto glycinergic neurotransmission strength remain speculative.

Our work shows that sustained SMO activation during 16 h in glycinergic neurons does not cause any effect on GlyT2 mRNA levels (Fig. 3) whereas its protein levels are dramatically affected during the same treatment (Fig. 1), suggesting that modulation of GlyT2 expression by the Hedgehog pathway does not occur at transcriptional or post-transcriptional levels. Thus, we explored how GlyT2 could be regulated at the protein level. We identified ubiquitination as the main mechanism driving SMO-mediated control of GlyT2, as PMM enhances GlyT2 ubiquitination levels and blockage of the E1 enzyme[63], required for the first step of the ubiquitination process, completely prevented the decrease in GlyT2 protein levels during SMO activation (Fig. 3).

We recently identified Ligand of Numb protein X1 and X2 (LNX1 and LNX2) as the first E3 ubiquitin ligases acting on GlyT2 (ref. [24]), although out of these two only LNX2 is expressed in glycinergic neurons[24,91]. We therefore tested whether LNX2 drives GlyT2 ubiquitination during Hedgehog activation. However, LNX2-deficient glycinergic neurons still showed Hedgehog-mediated degradation of GlyT2, indicating that this process relies on a different E3 ubiquitin ligase (Fig. S1). This result may not be

surprising, as a key feature of the ubiquitination process is its robust degree of redundancy and multiplicity, which allows that each protein can be targeted by multiple E3s[92,93]. We also explored whether PKC is involved in Hedgehog-mediated control of GlyT2 since PKC activation is known to induce ubiquitination and degradation of GlyT2 (refs. [14,25]) and Shh activation induces degradation of the glutamate transporter GLT-1 via PKC in astrocytes[88]. However, we found that GlyT2 downregulation persists during Hedgehog activation in glycinergic neurons in which PKC was pharmacologically inhibited (Fig. S2), indicating that the molecular mechanism controlling GlyT2 differs from the one modulating GLT-1. The identification of the molecular pathway downstream of Hedgehog signaling controlling GlyT2 expression remains, therefore, unsolved. Further research is required to solve this unknown mechanism, which will help to better understand how Hedgehog can control the biology of glycinergic neurons.

Zebrafish express homologs for more than 70% of human genes and their molecular underpinnings controlling neurotransmission are conserved with higher vertebrates[94]. For example, properties of glycinergic biology in zebrafish mimic well the functions observed in mammals, as, for example, (1) GlyT1 zebrafish mutants present excessive glycinergic transmission due to the lack of glycine recapture[71,95], (2) loss of glycine receptor function results in an hyperekplexia-like phenotypes[96,97], and (3) GlyT2 can be used as a reliable marker for glycinergic neurons[40,41]. To explore whether Hedgehog control of GlyT2 is conserved in vivo, we treated dechorionated zebrafish embryos at 24 hpf with cyclopamine for 24 h, determining at 48 hpf a robust increase of GlyT2 expression measured by both western blot and immunofluorescence of zebrafish spinal cord (Fig. 4). These results are in agreement with those obtained in primary neurons, where cyclopamine administration blocked GlyT2 reduction promoted by PMM (Fig. 2a, b) and suggest that theHedgehog signaling pathway physiologically modulates GlyT2 levels during development. In zebrafish, key neurodevelopmental events occur during the first 120 hpf (5 dpf): neural tube closure takes place approximately at 16 hpf, neural proliferation between 12 and 84 hpf, and neural migration between 18 and 96 hpf[98,99]. Regarding the expression of glycinergic neurotransmission elements, GlyT2 mRNA appears at 20 hpf[40,100] and GlyR α1 subunit at 24 hpf[101]. In this work, we modulate the Hedgehog pathway between 24 and 48 hpf, a window of time important for the correct expression of GlyT2 and the differentiation of glycinergic neurons. We hypothesize that the Hedgehog pathway may be involved in the specification of spinal cord inhibitory neurons by reducing glycinergic transmission strength through limiting GlyT2 activity. Before 30 hpf in zebrafish, GlyT2 and the GABAergic marker glutamic acid decarboxylase (GAD) co-localize in the majority of spinal cord neurons that show either marker, and longer times are required for these neurons to become either glycinergic or GABAergic[40,41]. We speculate that Hedgehog activation could act as a set point that temporarily restricts the differentiation of these neurons into glycinergic neurons by reducing GlyT2 expression and limiting glycinergic transmission strength[102], as GlyT2 is necessary for establishing and maintaining the identity of a glycinergic synapse[3,35–37]. Whether these effects persist in adulthood requires future investigation, but our data indicate that the novel control mechanism identified here by which GlyT2 expression is robustly controlled by Hedgehog activation status is a mechanism evolutionarily conserved that occurs in vitro and in vivo in glycinergic neurons of different vertebrate species. We hope that advancing our understanding of the molecular control of GlyT2 will help to frame future studies into the molecular basis of human pathologies associated with alterations in glycinergic neurotransmission, such as hyperekplexia or chronic pain.

## Methods

**Materials**. Zebrafish embryos and Wistar rats were bred under standard conditions at the Centro de Biología Molecular Severo Ochoa (CBMSO) in accordance with procedures approved in the Directive 2010/63/EU of the European Union with approval of the Research Ethics Committee of the Universidad Autónoma de Madrid (Comité de Ética de la Investigación UAM, CEI-UAM). Antibodies against N terminus of GlyT2 and GlyT1 were generated in house[103–105] (working dilution: 1:1000), while the other primary antibodies used were α3NKA (1:500, Santa Cruz Biotechnology, sc-16052), CRMP5 (1:2000, Abcam, ab36203), Syntaxin 1A (1:1000, Synaptic Systems), anti-ubiquitin (1:200, P4D1, Santa Cruz, sc-8017), and anti-α-tubulin (1:2000, T-6074, Sigma-Aldrich). All chemicals used were from Sigma-Aldrich unless otherwise noticed. Neurobasal medium and B27 supplement were purchased from Invitrogen.

**Small hairping RNAs (shRNAs) and lentiviral particles generation**. shRNA sequences were inserted into pLKO cloning vector (a gift from David Root; Addgene plasmid #10878). shRNA lentiviral particles were generated by transfection of HEK293T with the pLKO vectors containing shRNA, the packaging plasmid psPAX2 and the envelope plasmid pMD2G.
   shRNA sequences:
   Scramble shRNA:
   5′-CCTAAGGTTAAGTCGCCCTCGCTCGAGCGAGGGCGACTTAACCTTAGG-3′
   LNX2 shRNA:
   5′-CCACTGATCAACATCGTCATT-3′

**Primary cultures of brainstem and spinal cord neurons and transduction**. Primary cultures of brainstem and spinal cord neurons were prepared as described previously[24]. Briefly, the brainstem and spinal cord of Wistar rat fetuses were obtained at the 16th day of gestation, and the tissue was then mechanically disaggregated in HBSS (Invitrogen) containing 0.25% trypsin (Invitrogen) and 4 mg/ml DNase. Cells were plated at a density of 500,000 cells/well in 12 well multiwell plates, and they were incubated for 4 h in DMEM containing 10% FCS, 10 mM glucose, 10 mM sodium pyruvate, 0.5 mM glutamine, 0.05 mg/ml gentamicin, 0.01% streptomycin, and 100 U/ml penicillin G. After 4 h this buffer was replaced with Neurobasal/B27 culture medium containing 0.5 mM glutamine (50:1 by volume: Invitrogen), and 2 days later cytosine arabinoside (1 μM) was added to inhibit further glial growth. Neuronal cultures were infected with lentiviruses at DIV 1. Lentivirus preparations were added directly to the culture medium and maintained for 24 h at 5% v/v. Neurons were then replaced in fresh medium and culture was continued until 12 DIV.

**Quantitative real-time PCR (qPCR)**. Total RNA was extracted from zebrafish and cultured rat brainstem and spinal cord neurons following the TRI Reagent isolation protocol (Sigma-Aldrich).
   cDNA was synthetized using the iScript cDNA Synthesis kit (Bio-Rad) and qPCR was performed using Fast SYBR Green Master Mix (Thermo Fisher Scientific) following the manufacturer's recommendations. Relative gene expression levels were quantified by the $2^{-\Delta\Delta Ct}$ method using glyceraldehyde-3-phosphate dehydrogenase (GAPDH) as housekeeping gene.
   Primer sequences:
   GAPDH (Rat):
   5′-TCCCATTCTTCCACCTTTGA-3′; 5′-ATGTAGGCCATGAGGTCCAC-3′
   GlyT2 (Rat):
   5′-CAGCCAGGCCAATAAGACAT-3′; 5′-CCACCTGATCTCACCAGGAT-3′
   LNX2 (Rat):
   5′-CCGTGTGCCAAGATGTAATG-3′; 5′-GGATCCAGTTTCACCCTCAA-3′
   GAPDH (Zebrafish):
   5′-CTGGTCATTGATGGTCATGC-3′; 5′-CACCACCCTTAATGTGAGCA-3′
   GlyT2 (Zebrafish):
   5′-TTGCACAGCCAAACTCAGTC-3′; 5′-ACGGCTGTCTCAACATTTCC-3′

**Pharmacological treatments**. Brainstem and spinal cord neurons were incubated with PMM and/or cyclopamine at different times and concentrations, as indicated thorough the text, or with the vehicle alone in B27 supplemented Neurobasal medium at 37 °C. Manually dechorionated 24 hpf zebrafish embryos were incubated with 100 μM cyclopamine diluted in E3 medium (5 mM NaCl, 0.17 mM KCl, 0.33 mM CaCl₂, 0.33 mM MgSO₄, and 0.1% methylene blue) or with the vehicle alone for 24 h at 28 °C.

**Protein extraction and western blotting**. Primary neurons were lysed in RIPA buffer and extracted proteins were mixed with Laemmli buffer to prepare them for electrophoresis. Zebrafish embryos were directly lysed in Laemmli buffer at 95 °C for 15 min. Proteins bands from lysates were resolved by SDS-PAGE using a 4% stacking gel and 6 or 7.5% resolving gels. Then, proteins were transferred to PVDF membranes with a semi-dry transfer system and visualized by ECL (Bio-Rad) detection. The protein bands obtained by ECL using film exposures in the linear range were imaged using a GS-900 calibrated imaging densitometer (Bio-Rad) and

quantified using Image Lab Software (Bio-Rad). Standard errors were calculated after densitometry from at least three separate experiments.

**Ubiquitination assay**. Primary neurons were washed twice with phosphate-buffered saline (PBS) at 4 °C, harvested using a buffer containing 50 mM Tris, 150 mM NaCl and 50 mM N-ethylmaleimide with protease inhibitors PMSF and Sigma protease inhibitor cocktail (Ubiquitination Buffer, UB) and analyzed for protein quantification. Equal amounts of protein samples were centrifuged and pellets were resuspended in 90 μl of UB. Ten microliter of 10% sodium dodecyl sulfate (SDS) were added and samples were incubated for 10 min at 95 °C to eliminate protein interactions. Then, SDS was diluted by the subsequent addition of 34 μl of UB containing 4% Triton X100 and 1 ml of UB containing 1% Triton X100. After 30 min on rotary shaking at 4 °C, lysates were precleared with 40 μl of Protein G agarose (Invitrogen) for 30 min at 4 °C and then were incubated overnight with anti-GlyT2. Next day, samples were incubated with protein G agarose for 1 h at room temperature followed by four washes with ice-cold lysis buffer and elution in 2× Laemmli sample buffer at 75 °C. Cell lysates were probed by western blot analysis with specific antibodies.

**[³H]-glycine transport assays**. GlyT2 transport activity was measured as described previously[22]. Briefly, cells were incubated in a solution containing an isotopic dilution of 2 μCi/ml [3H] glycine (1.6 TBq/mmol; PerkinElmer Life Sciences) in PBS, yielding a 10 μM final glycine concentration. To measure specific GlyT2 glycine accumulation, the transport assay was performed in the presence of the GlyT2 antagonist ALX1393 (0.4 μM) and in the presence of the GlyT1 antagonist NFPS (10 μM) as these cultures present endogenous GlyT1 activity that would contaminate the measurements of GlyT2-mediated glycine uptake. Transport activity is normalized to the protein concentration and time of the reaction and expressed in pmol of glycine per mg of protein per minute (pmol of Gly/mg of protein/min). The reactions were terminated after 6 min by aspiration, followed by washing with PBS.

**Immunofluorescence of zebrafish embryos**. Dechorionated 48 hpf zebrafish embryos were fixed overnight in 4% PFA at 4 °C, soaked in 30% sucrose for 2–4 h, and embedded in OCT cryostat embedding medium (Tissue Tek). After freezing, 10 μm spinal cord transversal sections were cut on a cryostat at −20 °C. Then, sections were washed three times with PBS containing 1% Tween 20, blocked for 2 h at 4 °C with 3% BSA and incubated overnight at 4 °C with anti-GlyT2 rabbit antibody (1:200) in block solution. Next day, sections were washed six times with PBS containing 1% Tween 20, blocked again, and incubated overnight at 4 °C with DAPI (1:2000) (Merck) and anti-rabbit AlexaFluor-488 (1:500) (Thermo Fisher) in block solution. Finally, sections were washed four times with PBS containing 1% Tween 20, mounted using ProLong Gold Antifade Mountant (Thermo Fisher) and stored at 4 °C before imaging using laser scanning confocal microscopy on an A1R+ microscope (Nikon). Sections from vehicle or cyclopamine-treated zebrafish embryos were imaged under identical imaging conditions and fluorescence was quantified using ImageJ software.

**Data analysis and statistics**. All data and statistical analyses were performed using GraphPad Prism (GraphPad Software). Kruskal–Wallis test was used to compare multiple groups, with subsequent Dunn's post hoc test to determine the significant differences between samples. Kolmogorov–Smirnov and Mann–Whitney U tests were used to compare two separate groups. P values are denoted through the text as follows: $*p < 0.05$; $**p < 0.01$; $***p < 0.001$; $****p < 0.0001$; $p < 0.05$ or lower values were considered significantly different when compared by one-way Kruskal–Wallis (Dunns's post hoc test) or Kolmogorov–Smirnov and Mann–Whitney U tests. Thorough the text, the Box whisker plots represent mean (+), median (line), 25–75 percentile (box), 10–90 percentile (whisker), 1–99 percentile (X), and min–max (−) ranges.

**Reporting summary**. Further information on research design is available in the Nature Research Reporting Summary linked to this article.

## Data availability

The data analyzed during this study are available with this paper as Supplementary Data 1 and original raw data are available upon reasonable request. We also show uncropped original blots in the Supplementary Material as Figure S4.

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

## Acknowledgements
This work was supported by grants from the Spanish 'Ministerio de Economía y Competitividad', grant number SAF2017-84235-R (AEI/FEDER, EU) to B.L.-C. and from Fundación Ramón Areces to B.L.-C. Additionally, Institutional grants from the Fundación Ramón Areces and Banco de Santander are given to the CBMSO. J.d.J.-S receives support from the Diane Barrière Chair on Synaptic Bioenergetics (ICM Paris Brain Institute). J.d.J.-S is supported by the 2019 ATIP-Avenir program (Inserm and CRNS, France) and an ERC Starting Grant (ERC-StG-852873, EU). We thank Dr. Fernando Martín-Belmonte (Centro de Biología Molecular Severo Ochoa [CBMSO]) for generously providing the zebrafish embryos and experimental advice.

## Author contributions
Conceived and designed the experiments: C.A. and J.d.J.-S. Performed the experiments: A.R.-M., E.N. and S.G.-L. Contributed with resources: A.A.V., D.D. and N.R. Analyzed the data: A.R.-M., E.N., B.L.-C. and J.d.J.-S. Wrote the original draft: A.R.-M., C.A. and J.d.J.-S. Edited and reviewed the manuscript: A.A.V., D.D., N.R., A.R.-M., E.N., B.L.-C. and J.d.J.-S.

## Competing interests
The authors declare no competing interests.
