## [Transparent Peer Review File · Communications Biology]

Referee expertise:

Referee #1: neuro, glycinergic synapse

Referee #2: molecular neuroscience and genetics, zebrafish

Reviewers' comments:

Reviewer #1 (Remarks to the Author):

In this manuscript, Rocha-Muñoz and coauthors report that the glycine transporter GlyT2 is regulated by the Hedgehog pathway. The authors found that this pathway regulates GlyT2 mRNA levels and GlyT2 ubiquitination status in brainstem and spinal cord neuronal cultures, thus modulating its transport activity. In zebrafish, inhibition of Smoothed (which is de-repressed upon Hedgehog activation) increased GlyT2 expression, suggestive of endogenous downregulation of GlyT2 expression by the Hedgehog pathway at the developmental stage that was tested. The results are novel, and point to a potentially interesting mechanism of regulation of GlyT2 expression, but the authors do not explore the biological relevance of this mechanism – in which physiological circumstances does it occur, and how does it impact glycinergic transmission?

The presented data is convincing, but very preliminary, as the functional significance of the regulation of GlyT2 levels by the Hedgehog pathway is not explored. Thus, in the present form the study is unlikely to influence thinking in the field. It is important that the authors advance the work further, for example by testing whether regulating endogenous Hedgehog signaling affects glycinergic transmission in spinal cord neurons. In the discussion, the authors propose that Shh signaling could modulate GlyT2 expression in glycinergic neurons in the spinal cord to control the timing of glycinergic transmission. It would be very relevant to test this possibility.

Another aspects that could be further explored is how Hedgehog pathway is affecting GlyT2 ubiquitination. The authors made some efforts in this direction, but their results did not lead to the identification of the molecular pathway downstream of Hedgehog signaling. This is another aspect of the work that deserves further attention.

Reviewer #2 (Remarks to the Author):

Manuscript COMMSBIO-20-1387 by Rocha-Munoz and colleagues presents novel findings describing the regulation of the presynaptic glycine transporter GlyT2 by the Hedgehog signalling pathway, utilising both in vivo and in vivo models (rodent primary spinal cord neurones and zebrafish). Overall, the results of the study linking hedgehog signalling to control of GlyT2 levels are convincing. The authors show that: 1. Activation of hedgehog signalling using the small molecule agonist purmorphamine (PMM), which binds to and activates the GPCR Smoothed (SMO), causes a dose- and time-dependent decrease in GlyT2 expression levels in rodent primary spinal cord neurones; 2. This effect can be reversed by blockade of SMO with the antagonist cyclopamine (CYC); 3. The likely mechanism of GlyT2 down-regulation is via increased ubiquitination (but is not mediated by LNX2 or PKC signalling pathways); 4. Hedgehog inhibition in vivo increases GlyT2 expression, suggesting that hedgehog pathway activation serves to limit or regulate glycinergic activity during embryonic development. Overall, the conclusions are novel and likely to be of interest to others in the fields of glycine transporter and synaptic biology. The statistical analysis is appropriate and there is sufficient detail present for others reproduce the work.

While the results presented are convincing for the most part, further evidence is required to validate

some of the main conclusions of the study:

1. Figure 1D. Is the 30% reduction in detectable GlyT2 mRNA observed in rodent primary spinal cord neurones reproducible? The samples numbers here are low (n=3), is this three samples, or PCR in triplicate? Was the effect greater at 24 hrs? I also note that other synaptic proteins were assayed at different time points, 8 and 24 hrs (Fig. 1E). Due to the different time-points assayed, and the the absence of quantification, it cannot be stated that the levels of these other synaptic proteins were unaltered by purmorphamine treatment. I would also suggest that purmorphamine is tested in the in vivo zebrafish system, since this would show reproducibility of the effect in a different system. This assay should be straightforward, since the authors demonstrated that the hedgehog antagonist cyclopamine (CYC) had no apparent effect on GlyT2 transcript levels in zebrafish (Fig. 4).

2. For figure 3, I am struggling to understand the ubiquitination assay. It would appear that panel A shows immunoprecipitated GlyT2, visualised using an anti-ubiquitin antibody. However, I cannot see any discrete bands corresponding to GlyT2. Probing of the blots with an antibody against GlyT2 would appear to show a prominent protein at 75 kDa – but this cannot be seen at all in panel A. The cropping shown in the uncropped blots (in supplementary data) would also seem to exclude the 75kDa band. If a band corresponding to ubiquitinated GlyT2 cannot be visualised in panel A, then how can it be quantified relative to total GlyT2? This experiment is not at all convincing as it is presented, and casts a major doubt one of the major conclusions reached, i.e. that hedgehog is directly altering GlyT2 ubiquitination. Having said this, the experiments with PYR-41, a cell permeable inhibitor of ubiquitin-activating enzyme E1 (Fig. 3C,D) do support the involvement of E1 ubiquitin-activating enzymes in GlyT2 down-regulation. However, whether this effect is direct (via GlyT2 ubiquitination) or indirect (i.e. affecting a secondary protein that in turn controls GlyT2 levels) remains unclear from the data presented.

3. The zebrafish in vivo model would be a fantastic system to test the effects of different hedgehog ligands on endogenous GlyT2 levels. It is perhaps worth noting that the zebrafish genome contains five reported hedgehog ligands and two patched receptors, including duplicated copies of Shh (shha = shh and shhb = twhh). Have the authors examined the reported expression profiles of these genes to predict which might control zebrafish GlyT2 levels? Have any hedgehog ligands (e.g. purmorphamine) been tested in the zebrafish in vivo model to see if they result in GlyT2 down-regulation?

Professor Robert J Harvey, University of the Sunshine Coast, Australia

Reviewer #1 (Remarks to the Author):

In this manuscript, Rocha-Muñoz and coauthors report that the glycine transporter GlyT2 is regulated by the Hedgehog pathway. The authors found that this pathway regulates GlyT2 mRNA levels and GlyT2 ubiquitination status in brainstem and spinal cord neuronal cultures, thus modulating its transport activity. In zebrafish, inhibition of Smoothed (which is de-repressed upon Hedgehog activation) increased GlyT2 expression, suggestive of endogenous downregulation of GlyT2 expression by the Hedgehog pathway at the developmental stage that was tested. The results are novel, and point to a potentially interesting mechanism of regulation of GlyT2 expression, but the authors do not explore the biological relevance of this mechanism – in which physiological circumstances does it occur, and how does it impact glycinergic transmission?

The presented data is convincing, but very preliminary, as the functional significance of the regulation of GlyT2 levels by the Hedgehog pathway is not explored. Thus, in the present form the study is unlikely to influence thinking in the field. It is important that the authors advance the work further, for example by testing whether regulating endogenous Hedgehog signaling affects glycinergic transmission in spinal cord neurons. In the discussion, the authors propose that Shh signaling could modulate GlyT2 expression in glycinergic neurons in the spinal cord to control the timing of glycinergic transmission. It would be very relevant to test this possibility.

Another aspects that could be further explored is how Hedgehog pathway is affecting GlyT2 ubiquitination. The authors made some efforts in this direction, but their results did not lead to the identification of the molecular pathway downstream of Hedgehog signaling. This is another aspect of the work that deserves further attention.

We thank the reviewer for these comments and we agree that the presented data is novel and convincing, and supports well the claims made in the manuscript.

We strongly agree that the directions proposed are very interesting and, in fact, necessary to understand the impact of the Hedgehog pathway in regulating glycinergic transmission and controlling the timing of establishment of the identity glycinergic neurons during development. Additionally, we agree that dissecting and describing comprehensively the molecular pathway downstream of Hedgehog signaling would complement the details of the molecular link identified in this work on how Shh signaling modulates GlyT2 expression in glycinergic neurons.

However, we think the amount of work proposed by the referee would require years to be undertaken and we believe it is out of the scope of the current manuscript to solve the proposed experiments, despite their significance. We are confident that this work in its current form and scope provides a novel and important piece of data to understand the molecular underpinnings controlling the biology of glycinergic neurons. We intentionally submitted a short communication to Communications Biology to focus on providing the novelty of identifying this molecular link for the first time, while also presenting robust evidence supporting this finding. We are happy to hear that the referee considers that experiments shown in the manuscript are convincing and support our current claims. We believe our results open several fascinating avenues for future research, such as those suggested by the referee, and therefore we think the current scope and form of the manuscript will be of great interest to cellular and developmental neuroscientists and we feel it should not be considered preliminary just because it is not extensive in length.

Reviewer #2 (Remarks to the Author):

Manuscript COMMSBIO-20-1387 by Rocha-Munoz and colleagues presents novel findings describing the regulation of the presynaptic glycine transporter GlyT2 by the Hedgehog signalling pathway, utilising both *in vivo* and *in vivo* models (rodent primary spinal cord neurones and zebrafish). Overall, the results of the study linking hedgehog signalling to control of GlyT2 levels are convincing. The authors show that: 1. Activation of hedgehog signalling using the small molecule agonist purmorphamine (PMM), which binds to and activates the GPCR Smoothed (SMO), causes a dose- and time-dependent decrease in GlyT2 expression levels in rodent primary spinal cord neurones; 2. This effect can be reversed by blockade of SMO with the antagonist cyclopamine (CYC); 3. The likely mechanism of GlyT2 down-regulation is via increased ubiquitination (but is not mediated by LNX2 or PKC signalling pathways); 4. Hedgehog inhibition *in vivo* increases GlyT2 expression, suggesting that hedgehog pathway activation serves to limit or regulate glycinergic activity during embryonic development. Overall, the conclusions are novel and likely to be of interest to others in the fields of glycine transporter and synaptic biology. The statistical analysis is appropriate and there is sufficient detail present for others reproduce the work.

While the results presented are convincing for the most part, further evidence is required to validate some of the main conclusions of the study:

1. Figure 1D. Is the 30% reduction in detectable GlyT2 mRNA observed in rodent primary spinal cord neurones reproducible? The samples numbers here are low (n=3), is this three samples, or PCR in triplicate? Was the effect greater at 24 hrs?

The n=3 in Figure 1D refers to three PMM treatments in different samples. However, we agree with the referee that the number of qPCR experiments was low for the variability observed. We have repeated these experiments in order to increase the number of treated samples, including now a total of n=7 independent experiments. As the updated Figure 3A shows, once we include the new results, we do not see any significant reduction in GlyT2 mRNA levels after 16h of PMM treatment, which indicates that Hh signaling pathway is not acting on GlyT2 at mRNA level. Now these results agree with those obtained in zebrafish, where CYC treatment does not change GlyT2 mRNA levels. We are grateful to the reviewer for recognizing and pointing out this issue.

I also note that other synaptic proteins were assayed at different time points, 8 and 24 hrs (Fig. 1E). Due to the different time-points assayed, and the absence of quantification, it cannot be stated that the levels of these other synaptic proteins were unaltered by purmorphamine treatment.

Quantifications have been included in Figure, showing that other presynaptic proteins of glycinergic synapses remain unchanged.

I would also suggest that purmorphamine is tested in the *in vivo* zebrafish system, since this would show reproducibility of the effect in a different system. This assay should be

straightforward, since the authors demonstrated that the hedgehog antagonist cyclopamine (CYC) had no apparent effect on GlyT2 transcript levels in zebrafish (Fig. 4).

We agree with this suggestion and we have performed this experiment *in vivo* in zebrafish embryos, now included in the new version of the manuscript. However, we could not see a reduction on GlyT2 protein levels after treating with PMM (See new Fig. Sup. 3). This may not be surprising, as at this developmental stage, Shh signaling pathway is already robustly activated in zebrafish and pharmacological overstimulation on top may not be easily achievable. Of note, in primary neurons CYC treatments only show a conclusive effect in preventing the degradation of GlyT2 promoted by PMM (Fig. 2B, C) but do not drive a corresponding significant increase in GlyT2 levels, as conversely found in Zebrafish. This indicates that CYC is only effective when Smo is already activated, as occurring in zebrafish embryos, while PMM is only effective when Smo is not yet fully active, as found in rat primary neurons. Indeed, CYC, but not PMM, is usually used at this early stage of development to study Hh signaling pathway in zebrafish embryos (References 77-80). We have included new data in Sup. Fig 3 and the corresponding discussion in the text (end of page 6).

2. For figure 3, I am struggling to understand the ubiquitination assay. It would appear that panel A shows immunoprecipitated GlyT2, visualised using an anti-ubiquitin antibody. However, I cannot see any discrete bands corresponding to GlyT2. Probing of the blots with an antibody against GlyT2 would appear to show a prominent protein at 75 kDa – but this cannot be seen at all in panel A. The cropping shown in the uncropped blots (in supplementary data) would also seem to exclude the 75kDa band. If a band corresponding to ubiquitinated GlyT2 cannot be visualised in panel A, then how can it be quantified relative to total GlyT2? This experiment is not at all convincing as it is presented, and casts a major doubt one of the major conclusions reached, i.e. that hedgehog is directly altering GlyT2 ubiquitination. Having said this, the experiments with PYR-41, a cell permeable inhibitor of ubiquitin-activating enzyme E1 (Fig. 3C,D) do support the involvement of E1 ubiquitin-activating enzymes in GlyT2 down-regulation. However, whether this effect is direct (via GlyT2 ubiquitination) or indirect (i.e. affecting a secondary protein that in turn controls GlyT2 levels) remains unclear from the data presented.

In these ubiquitination assays, GlyT2 is immunoprecipitated under very stringent conditions from samples incubated at 95 °C for 10 min in a buffer containing 1% sodium dodecyl sulfate (SDS). This treatment denatures all proteins, disassembling their native structure and completely disrupting non-covalent protein-protein interactions. However, given that ubiquitin molecules are covalently attached to GlyT2, this treatment does not affect the number of ubiquitin molecules bound to GlyT2, allowing to quantify GlyT2 ubiquitination by immunoblotting GlyT2 immunoprecipitates against ubiquitin.

The anti-ubiquitin antibody used here recognizes all forms of polyubiquitin chains, which are formed by variable chain lengths. The consequence of this is that in these experiments we will identify all immunoprecipitated GlyT2 molecules ubiquitinated by variable polyubiquitin lengths, which when visualized together by western blot generates an elongated band spanning molecular weights higher than GlyT2. Since the higher forms of ubiquitinated GlyT2 have more ubiquitin molecules attached, they are detected with this antibody to a greater extent due to the presence of more epitopes available for the antibody to bind, and therefore are highly polyubiquitinated GlyT2 species more easily visualized by western blot. A 75kDa band could be theoretically observed in these assays, although it would correspond to monoubiquitinated GlyT2 forms; given the bias of the assay to identify highly ubiquitinated

forms, a 75KDa band could be visualized only if the majority of ubiquitinated GlyT2 species were monoubiquitinated. Although the fraction of monoubiquitinated GlyT2 species has not been determined yet in the field, the work presented here agrees with previous work from our lab using similar approaches (de la Rocha-Muñoz et al., 2021; de la Rocha-Muñoz et al., 2019; Villarejo-López et al., 2017; Arribas-González et al., 2015). Other groups studying polyubiquitination of other proteins find this type of results as well, which are accepted in the field as measurements of polyubiquitination (Grossman, S. R. et al (2003) *Science*, 300(5617), 342-344; Skieterska, K. et al (2015). *Dopamine Receptor Technologies* (pp. 139-156); Ovando-Roche, P. et al (2014). *Stem Cells*, 32(8), 2111-2122).

Despite a 75KDa form of GlyT2 is not detected with anti-ubiquitin antibodies in GlyT2 immunoprecipitates, we can quantify ubiquitination levels relative to total GlyT2 because our protocol allow us to immunoprecipitate GlyT2 alone, separated from its interactors due to the stringent conditions mentioned. To clarify this further, we have now included an extended explanation in the main text that describes more comprehensively how these experiments were performed.

3. The zebrafish in vivo model would be a fantastic system to test the effects of different hedgehog ligands on endogenous GlyT2 levels. It is perhaps worth noting that the zebrafish genome contains five reported hedgehog ligands and two patched receptors, including duplicated copies of Shh (shha = shh and shhb = twhh). Have the authors examined the reported expression profiles of these genes to predict which might control zebrafish GlyT2 levels? Have any hedgehog ligands (e.g. purmorphamine) been tested in the zebrafish in vivo model to see if they result in GlyT2 down-regulation?

We agree with the referee that zebrafish embryos are a great system to analyze the potential modulation of GlyT2 by different Hedgehog ligands during development. In this work, we did not explore the expression of these ligands in the zebrafish neural tube and we cannot conclude which one may regulate GlyT2 in vivo, although this will be an interesting future direction to better understand the process described in this paper. As far as we know, it has not been yet described how the Hh signaling pathway may control the development of glycinergic neurons in zebrafish. In zebrafish, the ligand Dhh is expressed in sertoli cells and is involved in testicular development (Shimizu et al., 2018), which does not make it a promising candidate to modulate GlyT2. Regarding indian hedgehog genes, lhha and lhhb (previously known as echidna hedgehog), both are expressed in notochord and the latter is required for the specification of oligodendrocyte progenitor cells from motor neuron precursors (Avaron et al., 2006; Chung et al., 2013). However, its role in neuron and interneuron differentiation is not described. From this family, the main regulators of neural tube development and those most likely involved in GlyT2 modulation are the Shh genes, Shha and Twhh. For example, these genes display redundant functions during diencephalic patterning (Scholpp et al., 2005), cooperatively induce all branchiomotor neurons (Bingham et al., 2001) and induce and maintain olig2 expression, which is crucial for motor neuron and oligodendrocyte development in the pMN precursor domain of the spinal cord (Park et al., 2004). Taking this into account, one could speculate that Shha and Twhh are most likely the Hh ligands responsible for the modulation of GlyT2. However, we feel current evidence is not sufficient for suggesting this possible connection and we would prefer to not speculate on this direction in the main text of this article.

REVIEWERS' COMMENTS:

Reviewer #1 (Remarks to the Author):

The work presented in this manuscript is well-performed and provides convincing and original evidence that the expression levels, and consequently the activity, of the glycine transporter GlyT2 is regulated by the Hedgehog pathway. The study is of interest to the community. However, as referred in the initial review, the work should be further developed to address the functional consequences of this regulatory mechanism to glycinergic transmission, and also to identify the pathways downstream of Hedgehog signaling that lead to GlyT2 ubiquitination.

The authors argue that these experiments would take a long time, and that they would be out of the scope of a short communication in *Communications Biology*. Nevertheless, some attempt should be made in the directions pointed out. In the present form, it is my opinion that the study is too incipient to be published in *Communications Biology*.

Minor issue: Panel identification in figures is missing. In figure 1 the order of the first three panels in the figure is different from what is described in the Results section.

Reviewer #2 (Remarks to the Author):

Manuscript COMMSBIO-20-1387A by Rocha-Munoz and colleagues presents novel findings describing the regulation of the presynaptic glycine transporter GlyT2 by the Hedgehog signalling pathway, utilising both *in vivo* and *in vitro* models (rodent primary spinal cord neurones and zebrafish). I have now reviewed the revised manuscript and the responses to referees comments. The manuscript has been improved by: 1. Repeating qPCR experiments, which now show no change in GlyT2 mRNA levels and quantification of presynaptic marker expression; 2. Testing purmorphamine PMM in zebrafish embryos; 3. Providing more methodological detail and explanation of the GlyT2 ubiquitination assay. While the authors did not address point 3 regarding hedgehog ligands, I concede that any speculation on the identity of the ligands influencing GlyT2 expression may require extensive further experimentation.